# The Effect of Coach–Athlete Relationships on Motor Behaviour in College Athletes—Mediating Effects of Psychological Needs

**DOI:** 10.3390/bs14070579

**Published:** 2024-07-08

**Authors:** Rong Zhang, Yong-Taek Rhim

**Affiliations:** Department of Sport and Leisure Studies, Namseoul University, Cheonan-si 31020, Republic of Korea; 18329141210@163.com

**Keywords:** coach–athlete relationship, psychological needs, motor behaviour, college athletes

## Abstract

This study explored the effect of the coach–athlete relationship perceived by college athletes on athletic behaviour, examining the mediating effect of psychological needs. This study was conducted with 254 college athletes using questionnaires, and the research instruments included the Coach–Athlete Relationship Scale, the Psychological Needs Scale, and the Athletic Behaviour Scale. The results of the study were as follows: (1) the coach–athlete relationship perceived by student-athletes had a direct positive and significant effect on athletic behaviours (β = 0.268, *p* < 0.001, and direct effect = 0.0344); and (2) psychological needs had a significant mediating effect between the coach–athlete relationship and athletic behaviours (indirect effect = 0.2433), and the mediating effect percentage was 41.56%. The overall mediation effect value was 0.5854. The findings of the study emphasized that, by improving the coach–athlete relationship, it could help to improve the psychological needs of college athletes and, thus, promote their positive motor behaviours. In the coach–athlete relationship, the effect of closeness on the promotion of motor behaviour was particularly significant, in which special attention and emphasis should be given to the closeness between coaches and athletes in a practical implementation. Subsequently, coaches should focus their training on enhancing the coach–athlete relationship as a crucial part of training in order to perform well in competitions.

## 1. Introduction

### 1.1. Research Background

The coach–athlete relationship has been linked to a wide range of variables connected to performance and athletes’ wellbeing, such as affects, coping, and sports satisfaction [1]. The 3C is a prominent model examining the coach–athlete relationship and is mainly focused on the dynamic interrelations between the coach and athletes and their mutual influence [2]. However, the model was updated to the 3 + 1Cs, which also evaluated the perception between athletes and coaches [3]. In the field of sport, research related to athletic ability has focused extensively on athletes’ performance as well as coaches’ behaviours, while research on athletes’ psychological needs and collective behaviours remains relatively understudied [1]. The mutual understanding, trust, respect, support, co-operation, and communication between coaches and athletes are important factors leading to a successful performance [4]. Cheng pointed out in his research that many successful cases of coaches and athletes establishing good relationships have emerged in China’s sports sector, such as Haiping Sun and Xiang Liu, etc., and these cases of co-operation have demonstrated the spirit of sincere co-operation and tenacity, and have resulted in the glory of a successful sports career [5]. In sports, the relationship between coaches and athletes is complex, dynamic, and multi-dimensional, so it is necessary to choose an appropriate method to study the coach–athlete relationship [6]. Recently, a conceptual definition of the coach–athlete relationship proposed a model that includes three main factors: task-related instructional factors, affective and cognitive factors related to social psychology, and behavioural, psychological factors related to trust between the coach and the athlete [7]. In addition, Jowett [8] and Philippe [9] developed a 10-item scale (the Coach–Athlete Relationship Questionnaire; CART-Q) comprising three factors (engagement, closeness, and complementarity) by qualitatively exploring coaches’ and athletes’ emotions, thinking, and behaviours, and validated the structural validity of this scale in a real-world sport context in China. In the field of sport, researchers have highlighted the behaviours and types of coaches as determinants of children’s and adolescents’ participation in sport [10]. In particular, coaching behaviours have been identified as significant in terms of adolescents’ sports participation, persistence, and performance [11]. Athletes with positive perceptions of their relationship with their coach felt fulfilled in terms of their autonomy, sense of competence, and need for relatedness, and had higher levels of personal growth and lower levels of negative self-efficacy. This literature suggested that the coach–athlete relationship may foster athletes’ adaptive coping strategies in competition.

Psychological needs include autonomy: the autonomous desire to express oneself and pursue one’s own behaviour [12]; competence: the desire to use one’s abilities and act effectively in one’s environment [13]; and relatedness: the desire to build stable relationships by trusting others to enhance concepts such as a sense of interpersonal relationship and belonging [14]. Thus, fulfilling psychological needs is considered to be a predictor variable that has a significant impact on athletes’ psychological and physical health, growth, intrinsic motivation and optimal performance, and self-actualization [15]. Previous research has reported that the relationship between the coaching relationship and athlete behaviour is mediated by autonomy, a sense of competence, and relatedness [16]. That is, the coaching relationship influences these three basic needs, and, ultimately, these needs influence athletes’ motor behaviour. Motor behaviour is based on the self-determination theory, in a sport context, about individual actions, thoughts, and body movements in the sport itself or in a sport scenario. In recent years, Maegu and Vallerand [17] emphasized that the relationship with the coach has a greater impact on athletes’ behaviour. In addition, social and situational support regarding psychological needs is closely related to the motivational stimulation and psychological growth of athletes [18]. To date, only a few studies examined the impact of the CAR on athletes’ affects [19]. This is particularly rare due to the possible connection between these variables and the impact of affective states on sports performance [20] and the increase in the literature on emotional contagion and interpersonal emotional regulation. Nevertheless, previous studies from other theoretical backgrounds close to CAR (e.g., coach behaviours and coach leadership) have examined this relationship. Particularly, previous works [21] revealed that warm and supportive coaches were significantly related to athletes’ positive affects. Otherwise, unsupportive coaches were significantly related to athletes’ negative affects (NAs) [22]. It has been shown that psychological needs (autonomy, sense of competence, and relatedness) play an important mediating role between the relationship with the coach and athletes’ behaviour [23]. Recent research has elucidated the relationship between coaches and athletes, pointing to strong links between athlete satisfaction and coaching behaviour, coach–athlete co-ordination, and perceptions of communication [24]. In particular, athletes with positive perceptions of and satisfaction with their relationship with their coach will strive harder for performance enhancement and group environment shaping [25], and also exhibit higher team cohesion and smooth communication [26].

### 1.2. The Present Study

The present study aimed to examine whether the satisfaction of psychological needs within the coach–athlete relational context serves as a mechanism by which the quality of coach–athlete relationships influences motor behaviour in college athletes. Individuals, including athletes, develop and refine their motor behaviours through interactions with key figures such as coaches, who are pivotal in shaping an athlete’s development and performance. Therefore, the study focused on the coach–athlete relationship, measured from a holistic perspective to capture its impact on athletes’ motor behaviour during training and competition. Based on previous research within the broader field of sports psychology, the hypothesis is formulated that the quality of the coach–athlete relationship affects the psychological needs of the athlete, which, in turn, influences the athlete’s motor behaviour. In particular, the coach–athlete relationship is expected to contribute to the athletes’ performance and personal and social development, thereby enhancing our understanding and accumulation of relevant knowledge. Therefore, this study will examine the effects of the coach–athlete relationship on college athletes’ motor behaviour and test the mediating role of psychological needs. Based on the previous literature [27], we hypothesized the following: (a) the coach–athlete relationship perceived by student-athletes had a direct positive and significant effect on athletic behaviours; (b) psychological needs had a mediating effect between the coach–athlete relationship and athletic behaviours; and, (c) by improving the coach–athlete relationship, it may help to improve the psychological needs of college athletes.

## 2. Research Objects and Methods

### 2.1. Participants and Procedure

Prior to data collection, our research team reviewed the questionnaire with three sport psychology PhDs. We compiled a list of athletic department directors to be surveyed and communicated with them via telephone or in-person visits to explain the study’s purpose and request assistance with data collection. Based on each school’s survey schedule, we provided participating athletes with a detailed explanation of the study’s purpose and instructions for completing the questionnaire, which was distributed via QR code. We adhered to the relevant Research Ethics Committee guidelines for projects involving human participants, and all participants provided informed consent. Using a convenience sampling method, we selected seven colleges and universities from Zhejiang Province and Anhui Province, surveying a total of 260 college athletes. After excluding six invalid questionnaires, we analyzed 254 valid questionnaires, resulting in a validity rate of 97.69%. The sample included 113 female students (44.49%) and 141 male students (55.51%); 77 students specializing in football (30.31%), 55 in track and field (21.65%), and 26 in basketball (10.24%); and 36 freshmen (14.17%), 42 sophomores (16.54%), 22 juniors (8.66%), and 154 seniors (60.63%) (Table 1).

### 2.2. Measurement Tools

#### 2.2.1. Coach–Athlete Relationship

In order to measure the coach–athlete relationship, we used Jowett and Ntoumanis’ (2004) Coach–Athlete Relationship Questionnaire (CART-Q) [28], which contains three factors (dedication, closeness, and complementarity) with 10 questions, and was evaluated using a 7-point Likert scale (1 = not at all, 7 = very much). The results of the analysis showed that the Coach–Athlete Relationship Questionnaire consisted of three factors and nine questions, of which the seventh question in the closeness factor was deleted because it did not fit conceptually. The overall Cronbach’s alpha coefficient of the Coach–Athlete Relationship Formal Questionnaire was 0.943 (>0.9), and the three factor Cronbach’s alpha coefficients were in the range of 0.815–0.959, implying that the internal consistency of the scales, both as a whole and in the dimensions, is very high.

The factors were extracted by the principal component method and rotated by the maximum variance method, and the coefficients were displayed with the absolute value set to 0.5. The cumulative variance explained was 87.393—items 8–10 belonged to component 1: complementarity, with a variance explained of 31.256 percent; items 4–6 belonged to component 2: closeness, with a variance explained of 28.244 percent; and items 1–3 belonged to component 3: dedication, with a variance explained of 27.244 percent, with a variance explained rate of 27.893%; and all question loadings were higher than 0.5, indicating that all passed the validity test. The factor loadings of the Coach–Athlete Relationship Questionnaire varied between a minimum of 0.615 and a maximum of 0.875, and the Kaiser–Mayer–Olkin fitness coefficient used to assess the suitability of the data for factor analyses was 0.864, which indicated potential correlation between the extracted latent variables, and the *p*-value of the Bartlett test of sphericity of <0.001 supported this view. These exploratory results were confirmed by the structure of confirmatory factor analysis and the fitness indices were Q: 3.117, TLI: 0.815, GFI: 0.945, CFI: 0.889, SRMR: 0.057, and NFI: 0.961. Thus, except for the Q value and SRMR, all the other indices exceeded the criterion values, indicating that the model is acceptable.

#### 2.2.2. Psychological Needs

In order to measure the level of psychological needs of college athletes, we used the Psychological Needs of Athletes Questionnaire developed by Park and Kim (2008) based on self-determination theory [29]. The questionnaire contains three factors with 12 questions (autonomy, sense of competence, and relatedness), and was evaluated using a 5-point Likert scale (1 point = not at all, 5 points = very much). The results of the analysis showed that the psychological needs questionnaire had a rotated component matrix with one and only one factor loading value greater than 0.5 for all variables, where the factor loading values for each factor ranged from 0.569 to 0.934, respectively, with high reliability, and all the question items were retained, which resulted in three factors and 12 questions, with reliability analyses ranging from 0.820 to 0.948. The questionnaire Cronbach’s alpha coefficient was 0.853, the Psychological Needs Questionnaire explained 76.409% of the variance overall, the Kaiser–Mayer–Olkin fitness coefficient was 0.797, and the *p*-value of Bartlett’s test of sphericity was less than 0.05, and these results support the applicability of factor analysis. The results of confirmatory factor analysis also showed good fitness with fitness indices of Q: 2.966, GFI: 0.959, CFI: 0.972, RMR: 0.049, and NFI: 0.951, and all indicators except RMR exceeded the standard values, indicating that the model is acceptable.

The results of the analysis showed that the psychological needs questionnaire had a rotated component matrix with one and only one factor loading value greater than 0.5 for all variables, where the factor loading values for each factor ranged from 0.569 to 0.934, respectively, with high reliability, and all the question items were retained, which resulted in three factors and 12 questions, with reliability analyses ranging from 0.820 to 0.948. The questionnaire Cronbach’s alpha coefficient was 0.853, the Psychological Needs Questionnaire explained 76.409% of the variance overall, the Kaiser–Mayer–Olkin fitness coefficient was 0.797, and the *p*-value of Bartlett’s test of sphericity was less than 0.05, and these results support the applicability of factor analysis. The results of confirmatory factor analysis also showed good fitness with fitness indices of Q: 2.966, GFI: 0.959, CFI: 0.972, RMR: 0.049, and NFI: 0.951, and all indicators except RMR exceeded the standard values, indicating that the model is acceptable.

#### 2.2.3. Motor Behaviour

This study used Wilson and Rogers’ motor behaviour questionnaire (Behavioural Exercise Questionnaire-2: BREQ-2), which was modified by Zhang [30]. This questionnaire included four questions on unmotivated behaviour, four questions on external regulation, three questions on internal regulation, four questions on confirmatory regulation, and four questions on internal regulation, totalling five factors and nineteen questions. The questionnaire was based on a 5-point Likert scale, asking respondents to choose between “not at all (1 point)” and “very much (5 points)”: the Measure of Exercise Regulation—BREQ-2 The BREQ-2 is used to measure the underlying motivational regulation of students in relation to exercise participation. It consists of five subscales, (1) intrinsic; (2) identified; (3) introjected; (4) external; and (5) amotivation, and is based on individual subscale items. The BREQ-2 has been shown to have excellent construct validity and reliability. The change in alpha coefficients was analyzed by reliability analysis and the items with low correlation coefficients were deleted, and the final Motor Behaviour Questionnaire consisted of 23 questions and three factors, “Intrinsic”, “Discriminative”, and “Extrinsic”, with a Cronbach’s coefficient of 0.958 and a Kaiser–Mayer–Olkin of 0.844, and factor loadings for each sub-factor ranging from 0.551 to 0.884, which explained a total of 78.439% of the variance, with *p*-values of Bartlett’s Test of Sphericity for the approximation of the chi-square of 8972.611 and *p* < 0.001, showing the adaptability of the model. Confirmatory factor analysis of the fitness results was as follows—Q: 2.178, GFI: 0.931, CFI: 0.959, RMR: 0.048, and NFI: 0.972, indicating that the model was acceptable.

## 3. Research Results

### 3.1. Primary Analysis

A confirmatory factor analysis (CFA) was conducted to validate the factor structure of the questionnaire used in this study. Prior to the analysis, the normality of the sample distribution was assessed using a P-P plot, which indicated a distribution close to a straight line, with a skewness of up to 0.901 and kurtosis of up to 0.887, suggesting the adherence to normality assumptions. The factor estimation employed the maximum likelihood method. A CFA was then performed on both a one-factor and a three-factor structure model. The fit indices for each model were compared (see Table 2). Model 1, representing a single-factor structure, showed χ^2^/df = 4.922, CFI = 0.683, TLI = 0.667, SRMR = 0.103, and RMSEA = 0.109. Model 2, representing a three-factor structure, demonstrated an improved fit with χ^2^/df = 2.791, CFI = 0.856, TLI = 0.849, SRMR = 0.090, and RMSEA = 0.074. The significance level is indicated by *** *p* < 0.001. These results underscore the adequacy of both the single-factor and three-factor models for interpreting the original questionnaire. This analysis provides a robust validation of the factor structure essential for the subsequent statistical and substantive interpretations in this study.

The collected data underwent rigorous analysis using the SPSS 25.0 and MPLUS 8.0 software, aligning with the study’s objectives. Initially, an exploratory factor analysis (EFA) and internal consistency analysis were employed, followed by a confirmatory factor analysis (CFA), to establish the validity and reliability of each questionnaire’s factor structure. Normality testing of the sample distribution preceded the factor analysis, confirming that the data approximated a normal distribution. The maximum likelihood method was utilized for factor estimation. The CFA was subsequently applied to evaluate a three-factor structure model, which demonstrated a satisfactory fit based on the analysis results. Furthermore, descriptive statistics and correlation analyses were conducted to scrutinize the sub-factors derived from the validated measurement variables identified through factor analysis. To explore the influence of these variables, stepwise multiple regression analyses were performed. Lastly, path analyses and mediation effects tests were executed to validate the hypothesized relationships within the research model. These comprehensive analytical steps were crucial for assessing the data and interpreting findings within the context of the study’s research objectives.

### 3.2. Descriptive Statistics and Correlations

The mean and standard deviation of the measured coach–athlete relationship, psychological needs, and motor behaviour sub-factors were calculated in this study. Table 3 presents the results of the descriptive statistical analyses of these sub-factors. The analyses indicated that respondents perceived the highest levels of complementarity and closeness but lower levels of dedication in their relationships with their coaches. Regarding psychological needs, relationship needs were higher than autonomy needs. “Coach–athlete relationships”, “Intrinsic”, and “Motor Behaviour” exhibited high variances and larger SEM values, indicating greater variability. In contrast, variables such as “Complementarity”, “Autonomy”, and “Sense of capability” showed lower variances and smaller SEM values, suggesting less variability and more precise estimates. Finally, for motor behaviour, intrinsic motivation was the highest, while extrinsic motivation levels were the lowest, indicating overall higher levels of intrinsic motivation.

Bivariate correlation is a widely used statistical method to measure the linear relationship between two variables. The method helps to uncover the interdependencies between variables and can be evaluated for significance through hypothesis testing. Based on the analysis of the correlations between the sub-factors of each factor, the results are shown in Table 4 below. The results of the analysis indicate several significant correlations. There is a notably strong correlation between coach–athlete relationships and psychological needs (r = 0.551 ***). Additionally, sub-factors of the coach–athlete relationship, such as intimacy (r = 0.932 ***), complementarity (r = 0.909 ***), and dedication (r = 0.906 ***), also show significant correlations with psychological needs and motor behaviour. Specifically, psychological needs are highly correlated with relevance (r = 0.845 ***), autonomy (r = 0.802 ***), and sense of capability (r = 0.773 ***). Motor behaviour exhibits strong correlations with intrinsic motivation (r = 0.992 ***), identified motivation (r = 0.921 ***), and external motivation (r = 0.645 ***). However, the correlation between internal control and psychological needs, as well as athletic behaviours, is primarily associated with closeness. These findings suggest a complex network of relationships among the variables, with certain variables such as coach–athlete relationships and psychological needs playing pivotal roles. The strong positive correlations indicate that enhancing coach–athlete relationships and addressing psychological needs could positively impact motor behaviour and intrinsic motivation. Based on these correlation analyses, a multiple regression analysis was executed to further explore these relationships within the research model.

### 3.3. Path Analysis

In this study, MPLUS 8.0 was used to conduct a validation factor analysis on the structural validation sample data (*n* = 254), which verified a good model fit of the coach–athlete relationship on sport behaviour, and a path analysis was conducted to estimate the path coefficients of the structural model and assess the mediating effect of psychological demands by SEM, and the results are shown in Table 5.

According to Yzerbyt in conjunction with the methodology of testing mediated models with moderation using the macro program PROCESS 3.5 of SPSS in Hayes [31], the parameters of the three regression equations need to be estimated. Equation (1) shows the effect of the relationship between the independent variable (coach–athlete relationship) and the dependent variable (motor behaviour); Equation (2) is used to estimate the effect of the relationship between the mediating variable (psychological needs) and the dependent variable (motor behaviour); and Equation (3) is used to estimate the mediating effect of the mediating variable (psychological needs) on the relationship between the independent variable (coach–athlete relationship) and the dependent variable (motor behaviour). All predictor variables were standardized in each equation [32], and then moderated mediated effects analyses were conducted controlling for gender and age, using a bias-corrected percentile Bootstrap method test with 5000 repetitions of the sample and calculating 95% confidence intervals, with all predictor variables having variance inflation factors no higher than 1.40, so there is no serious multicollinearity problem. The results are shown in Table 6.

In the first step, the simple mediation model was tested, and Model 3 was chosen to test the mediating role of psychological needs between the coach–athlete relationship and motor behaviour. A regression analysis showed that the coach–athlete relationship had a significant positive predictive effect on motor behaviour (β = 0.6457, *p* < 0.001); after psychological needs were included in the regression equation, the predictive effect of the coach–athlete relationship on motor behaviour remained significant (β = 0.2433, *p* < 0.001), the coach–athlete relationship positively predicted psychological needs (β = 0.4732, *p* < 0.001), and psychological needs positively predicted motor behaviour (β = 1.2371, *p* < 0.001), with Boot SE = 0.1647, with a 95% confidence interval of [0.9747, 1.4876], suggesting a significant mediating role of psychological needs between the coach–athlete relationship and motor behaviour.

In the second step, the mediating effect of psychological needs was tested. Regression analyses showed that the coach–athlete relationship positively predicted psychological needs (β = 0.3845, *t* = 6.8411, *p* < 0.001), but the positive predictive effect of psychological needs on the interaction term of the coach–athlete relationship and motor behaviour was significant (β = −0.0116, *t* = −2.3083, *p* < 0.05), with a 95% confidence interval of [−0.0214, −0.0017]. The regression equation for the moderated effect path was Y = 49.9093 + 0.3845 × X + 0.0621 × W − 0.0116 × XW, with *p* < 0.05, total effect = 0.5854, direct effect = 0.0344, and indirect effect = 0.2433, and the mediating effects were all significant, and the percentage of intermediate effect was 0.2433/0.5854 = 41.56%, with the 95% CI (0.3510, 0.8179); it can be assumed that psychological needs have a mediating role in the coach–athlete relationship on sports behaviour. The results of the model plot are shown in Figure 1. This result shows that psychological needs mediate the path of “coach–athlete relationship → motor behaviour”. 

## 4. Discussion

The main purpose of this study was to explore in depth the effects of the coach–athlete relationship on psychological needs and motor behaviour in a sport environment. The results of the study revealed the direct effect of the coach–athlete relationship on psychological needs and motor behaviour, and further analyses showed that psychological needs mediated the relationship between the coach–athlete relationship and motor behaviour, and that the mediating effect was positively significant, a finding that underscores the importance of the coach–athlete relationship in stimulating the psychological needs of the athletes and positively influencing motor behaviour.

### 4.1. Direct Effects of Coach–Athlete Relationships on Psychological Needs and Motor Behaviour

Firstly, athletes’ perceptions of the coach–athlete relationship were found to directly influence their psychological needs in a collegiate sport setting. We observed a causal relationship between the coach–athlete relationship and psychological needs, a finding that is consistent with previous research [33], which concluded that athletes’ psychological needs vary with the coach–athlete relationship. In addition, Papich [34] study noted that the coach–athlete fit affects the satisfaction of psychological needs, especially when athletes perceive that the coach’s training style and personality fit with them—athletes’ autonomy needs and relationship needs satisfaction are higher, which, to some extent, corroborates our findings.

Second, the causal relationship between athletes’ perceptions of autonomy and the coach–athlete relationship is crucial. The findings suggest that athletes who perceive the coach–athlete relationship positively have the highest satisfaction with autonomy needs. This indicates that autonomy needs are a significant predictor of intrinsic motivation. Furthermore, coaching behaviours influence athletes’ perceptions of autonomy, competence, and relational needs, ultimately affecting their motivation. However, a slight departure from Endo’s findings among taekwondo athletes revealed that intimacy and dedication factors influenced relational and autonomy needs [35,36], while the dedication factor affected competence needs [37]. These differences may be explained by the variations between individual and group sports programs.

By examining the relationship between the coach–athlete relationship and motor behaviour, we found that the coach–athlete relationship influences motor behaviour. This result is partially consistent with the findings of previous research related to the coach–athlete relationship and motor behaviour [38]. Specifically, the subcomponents of the intimacy, dedication, and complementarity of the coach–athlete relationship all positively influenced intrinsic regulation and extrinsic regulation. Additionally, prior research has confirmed that athletes’ perceptions of the coach–athlete relationship have a significant impact on teamwork and team efficacy, a finding that was also supported in our study [39]. Factors such as intimacy, dedication, and complementarity positively influence athletes’ motivation and motor behaviour in a university sport setting. Several previous studies [14,40,41,42,43] indicated that these factors in the coach–athlete relationship can motivate athletes’ willingness to participate in sport. Furthermore, according to the Ameloot self-determination theory [44], there is a correlation between intrinsic motivation and extrinsic regulation in the coach–athlete relationship and athlete satisfaction, with intrinsic motivation positively affecting satisfaction and extrinsic regulation negatively affecting it. The findings of this study also emphasized the role of intimacy as a significant predictor of intrinsic regulation.

### 4.2. Mediating Effects of Psychological Needs between Coach–Athlete Relationship and Motor Behaviour

Based on an in-depth analysis of the relationship between psychological needs and motor behaviour, we found a positive and significant direct effect of psychological needs on motor behaviour. This finding is consistent with the findings of numerous previous studies [45], which emphasized the need for coaches to not only guide athletes technically and tactically, but also put in hard work in caring for athletes’ lives and guiding athletes’ career development. In the “coach–athlete” relationship, the relational contribution of coaches is crucial to the growth of athletes [46]. Studying the relationship between psychological needs and athletic behaviours of college athletes, Yoo found that, when athletes felt that their autonomy needs were met and the coach–athlete relationship was good, there was a positive trend in athletic motivation, and athletic behaviours were more positive [47]. Through a comprehensive examination of the causal relationship between the coach–athlete relationship, psychological needs, and motor behaviours, the results indicate that there is a significant direct effect of the coach–athlete relationship on the regulation of psychological needs and motor behaviours. At the same time, we found that psychological needs played a mediating role between the coach–athlete relationship and motor behaviour, with a mediating effect value of 41.56%. This result reveals the close connection between psychological needs and an individual’s social relationships, particularly the coach–athlete relationship. Psychological needs play a mediating role in the coach–athlete relationship by motivating more positive motor behaviours among college athletes. Psychological needs triggered through the coach–athlete relationship are more effective in promoting positive motor behaviours. However, the influence of psychological needs on motor behaviour may be somewhat limited by the prerequisites of a good coach–athlete relationship. Therefore, the impact of psychological needs on motor behaviour is likely more profound when a strong coach–athlete relationship is present [48]. The coach–athlete relationship, as an affective experience, helps maintain stability and meaningful social connections, rebuilds individuals’ confidence in their social goals, develops a sense of belonging and security [49], and, thus, increases psychological needs [50], leading to more active participation in motor behaviours among college athletes.

## 5. Conclusions

The purpose of this study was to explore the associations between the coach–athlete relationships, psychological needs, and motor behaviours of college athletes. The study revealed the following conclusions: (1) the coach–athlete relationship significantly positively affects psychological needs and athletic behaviour; (2) in the context of the coach–athlete relationship, closeness had the most significant effect on motor behaviour, followed by complementarity and dedication; and (3) psychological needs mediate the relationship between the coach–athlete relationship and motor behaviour, emphasizing their role as a bridge connecting these variables. A strong coach–athlete relationship enhances athletes’ psychological needs satisfaction, thereby promoting more active participation in motor behaviours. Future research should further explore the mechanisms by which different types of coach–athlete relationships affect psychological needs and motor behaviours, and design appropriate interventions to promote good relationship building and the positive development of motor behaviours. These findings have important theoretical and practical implications for fostering a healthy and positive sports culture and improving overall athlete performance.

## Figures and Tables

**Figure 1 behavsci-14-00579-f001:**
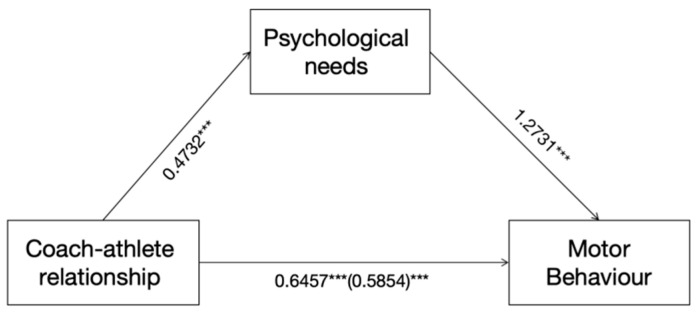
Diagram of the intermediary model. *** *p* < 0.001.

**Table 1 behavsci-14-00579-t001:** Demographic information of the participants (*n* = 254).

Variable		Number	Percent
Sex	Male	141	55.51%
	Female	113	44.49%
Grades	Freshman	36	14.17%
	Sophomore	42	16.54%
	Junior	22	8.66%
	Senior	154	60.63%
Sports event	Soccer	77	30.31%
	Basketball	26	10.24%
	Volleyball	17	6.69%
	Track and field	55	21.65%
	Tennis balls	20	7.87%
	Others	59	23.23%
Major	Physical education	177	69.69%
	Non-physical education	77	30.31%

**Table 2 behavsci-14-00579-t002:** Results of the confirmatory factor analysis.

	χ^2^/df	CFI	TLI	SRMR	RMESA
Model 1	4.922 ***	0.683	0.667	0.103	0.109
Model 2	2.791 ***	0.856	0.849	0.090	0.074

Model 1 is a one-factor model and Model 2 is a three-factor model. *** *p* < 0.001.

**Table 3 behavsci-14-00579-t003:** Descriptive statistics of measured variables (*n* = 254).

Variable	M	SD	Kurtosis	Cronbach’s Alpha	Skewness
Intimacy	19.37	2.861	15.070	0.947	−3.307
Complementarity	19.32	2.156	1.964	0.951	−1.1278
Dedication	18.38	2.694	3.824	0.922	−1.711
Coach–athlete relationships	57.07	7.067	9.045	0.943	−2.448
Relevance	18.89	3.302	−0.881	0.836	0.183
Autonomy	12.46	2.023	0.370	0.887	−0.705
Sense of capability	18.38	2.255	5.175	0.851	−1.818
Psychological needs	49.73	6.065	2.363	0.853	−0.807
Intrinsic	69.30	9.084	6.020	0.951	−2.067
Identified	55.18	6.699	5.785	0.963	−1.844
External	15.33	3.764	−1.009	0.969	0.089
Motor behaviour	139.81	17.004	4.951	0.958	−1.583

**Table 4 behavsci-14-00579-t004:** Bivariate correlations of all study variables.

Variable	1	2	3	4	5	6	7	8	9	10	11	12
1. Intimacy	1											
2. Complementarity	0.800 ***	1										
3. Dedication	0.743 ***	0.735 ***	1									
4. Coach–athlete relationships	0.932 ***	0.909 ***	0.906 ***	1								
5. Relevance	0.271 ***	0.273 ***	0.432 ***	0.357 ***	1							
6. Autonomy	0.271 ***	0.249 ***	0.141 *	0.239 ***	0.512 ***	1						
7. Sense of capability	0.738 ***	0.663 ***	0.641 ***	0.745 ***	0.351 ***	0.510 ***	1					
8. Psychological needs	0.512 ***	0.478 ***	0.520 ***	0.551 ***	0.845 ***	0.802 ***	0.773 ***	1				
9. Intrinsic	0.509 ***	0.414 ***	0.363 ***	0.471 ***	0.261 ***	0.470 ***	0.389 ***	0.443 ***	1			
10. Identified	0.446 ***	0.427 ***	0.383 ***	0.457 ***	0.458 ***	0.628 ***	0.487 ***	0.640 ***	0.758 ***	1		
11. External	0.269 ***	0.468 ***	0.292 ***	0.363 ***	0.428 ***	0.303 ***	0.319 ***	0.453 ***	0.401 ***	0.553 ***	1	
12. Motor behaviour	0.507 ***	0.493 ***	0.410 ***	0.512 ***	0.415 ***	0.566 ***	0.471 ***	0.589 ***	0.992 ***	0.921 ***	0.645 ***	1

* *p* < 0.05 and *** *p* < 0.001.

**Table 5 behavsci-14-00579-t005:** Path analysis and mediated effects results.

Path Relationship	β	SE	z	Bootstrap 95% CI		*p*
				Lower	Upper	
Coach–athlete relationships → Psychological needs	0.50	0.05	9.00	0.35	0.55	<0.001
Psychological needs → Motor behaviour	0.35	0.07	4.29	0.16	0.44	<0.001
Coach–athlete relationships → Motor behaviour	0.25	0.06	3.33	0.08	0.32	<0.001
Coach–athlete relationships → Psychological needs → Motor behaviour	0.175	0.04	3.50	0.06	0.22	<0.001

**Table 6 behavsci-14-00579-t006:** Coach–athlete relationship in relation to sports behaviour: moderated mediating effect (Group Environment).

Predictor Variable	Equation (1) Psychological Needs			Equation (2) Motor Behaviour			Equation (3) Psychological Needs		
	B	SE	*t*	B	SE	*t*	B	SE	*t*
Coach–athlete relationship	0.4732	0.0451	10.492 ***	0.2433	0.0521	4.5697 ***			
Psychological needs				1.2371	0.1647	7.5134 ***			
Coach–athlete relationship × Motor behaviour							−0.0116	0.0050	−2.3083 *
R^2^	0.3040			0.3974			0.3236		
F	100.0767			82.7565			39.8637		

* *p* < 0.05, and *** *p* < 0.001.

## Data Availability

The original contributions presented in the study are included in the article; further inquiries can be directed to the corresponding authors.

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
