# Peer review of "The Effect of Coach–Athlete Relationships on Motor Behaviour in College Athletes—Mediating Effects of Psychological Needs"

_behavsci, 2024, doi:10.3390/bs14070579_

Round 1

Reviewer 1 Report

Comments and Suggestions for Authors

Thanks for providing me the opportunity to review such an interesting article. Please, see in the attachment some minor issues to be handled.

Comments on the Quality of English Language

The quality of the english language is fine.

Author Response

Dear Reviewers:

Thank you for your letter and for the reviewers’ comments concerning our manuscript. All these comments are valuable and helpful in revising and improving our thesis, as well as providing important guidance for our research. We have carefully studied these comments and made revisions, which we hope will be accepted by you. Where changes have been made, we have highlighted them in yellow in the revised version.

We would love to thank you for allowing us to resubmit a revised copy of the manuscript and we highly appreciate your time and consideration.

Sincerely.

The main corrections in the paper and the responds to the reviewer’s comments are as flowing:

Reviewer :

Q1.Abstract:“this phrase does not add any extra value, it is too general”

Response:Thank you very much for your valuable comments.I have amended the sentence to:Subsequently, coaches should focus their training on enhancing coach-athlete relationship as a crucial part of training in order to perform in competitions.

Q2.Introduction:“this is not scientifically proved...”“engagement”“Here you have one of the latest articles about coach-athlete relationship: González-García, H., Martinent, G., & Nicolas, M. (2023). The mediating roles of pre-competitive coping and affective states in the relationships between coach-athlete relationship, satisfaction and attainment of achievement goals. International Journal of Sport and Exercise Psychology, 1–16. https://doi.org/10.1080/1612197X.2023.2190346”

Response:I have addressed your comments with the following changes:

For the first issue, I have supplemented my study with relevant research. For

the second issue, I have revised the wording as suggested. For the third issue, I carefully reviewed the papers you recommended, summarized their findings, and incorporated them into my paper with proper citations. Thank you for

your valuable feedback.

Cheng pointed out in his research: many successful cases of coaches and athletes establishing good relationships have emerged in China's sports sector, such as Haiping Sun and Xiang Liu, etc., and these cases of co-operation have demonstrated the spirit of sincere co-operation and tenacity, and have resulted in the glory of the sports career[5].

This literature suggested that coach-athlete relationship may foster athletes’ adaptive coping strategies in competition.

  1. CHENG Hongren, SUN Haiping, ZHONG Yaping & YUAN Tinggang. 2011 A new species of the genus Pseudourostyla (Hymenoptera, Braconidae) from China.  An analysis of Sun Haiping's hurdles training concept and practice - based on the perspective of individualised training. Journal of Shanghai Institute of Physical Education . 2023.(09), 52-63.
  2. González-García, H., Martinent, G., & Nicolas, M. (2023). The mediating roles of pre-competitive coping and affective states in the relationships between coach-athlete relationship, satisfaction and attainment of achievement goals. International Journal of Sport and Exercise Psychology, 1– https://doi.org/10.1080/1612197X.2023.2190346

Q3.Research Objects and Methods:“tis it reviewed by a ethics board?”“correlation is not so reliable to be used as an statistic”

Response: Thank you very much for your valuable comments. I have made changes based on the suggestions you made.

3.1 Descriptive statistic

The mean and standard deviation of the measured coach-athlete relationship, psychological needs, and motor behavior sub-factors were calculated in this study. Table 3 presents the results of the descriptive statistical analyses of these sub-factors. The analyses indicated that respondents perceived the highest levels of complementarity and closeness but lower levels of dedication in their relationships with their coaches. Regarding psychological needs, relationship needs were higher than autonomy needs. "Coach-Athlete Relationships," "Intrinsic," and "Motor Behavior" exhibited high variances and larger SEM values, indicating greater variability. In contrast, variables such as "Complementarity," "Autonomy," and "Sense of Capability" showed lower variances and smaller SEM values, suggesting less variability and more precise estimates. Finally, for motor behavior, intrinsic motivation was the highest, while extrinsic motivation levels were the lowest, indicating overall higher levels of intrinsic motivation.

3.2 Correlation analysis

Bivariate correlation is a widely used statistical method to measure the linear relationship between two variables.The methods help to uncover the interdependencies between variables and can be evaluated for significance through hypothesis testing. Based on the analysis of the correlations between the sub-factors of each factor, the results are shown in Table 4 below. The results of the analysis show that there is a significant correlation between coach-athlete relationship and psychological needs, while there is only a correlation with closeness between internal control and psychological needs and athletic behaviours, other than that, other correlations are not significant. In addition, the subfactors of coach-athlete relationship showed a positive correlation with psychological needs and a significant correlation with motor behaviour. Therefore, based on these correlation analyses, a multiple regression analysis based on the research model was executed.

Reviewer 2 Report

Comments and Suggestions for Authors

Thank you for the opportunities to review this manuscript. I believe it has a good topic, but several things need to revise. Introduction is generally well written, but the methods and results needs to revise. 

My recommendation to the Authors:

Introduction

·      More studies, aspects need in the introduction to show the main research gap. 

·      The study goal needs more precise 

·      I suggest making hypotheses. Especially the Authors used an empirical model to their results, which could be also used as a hypothesis. 

Methods

·      Consider moving procedure to the first subchapter. In this case the readers were understand the data collection in the beginning of the chapter. 

·      Furthermore, I suggest combining participants with procedure ("Participants and Procedure)

·      The measures are well detailed. However, I suggest to put your primary analysis to the results and only make clear descriptions in this study on the measures. 

·      As I understand the Authors used exploratory (EFA) and confirmatory factor analysis (CFA) in this study. However, these are well known measures, and it is unnecessary to use both in these cases (Usually EFA and CFA are conducted on different sample). So, I encourage to only use CFA and move them to the results. 

·      Statistical analysis must be more detailed. For example, what was the dependent variables on step wise regression; What kind of fit indices was used for CFA; etc.

Results

·      Line 191: should be "Descriptive statistic"

·      Table 4: Why were these measures used? M; SD; Skewness and Kurtosis could be enough, but reliability measures can be also added to this table. 

·      Correlation table seems to have missing values.

·      What was the model fit for figure 1, as it was mentioned it is a path analysis. 

Discussion, conclusion

·      This part needs to revise after revising the methods and the results. However the structure and main point of this part are well established. 

Comments on the Quality of English Language

English Language is fine

Author Response

Dear Reviewer:

Thank you for your letter and for the reviewers’ comments concerning our manuscript. All these comments are valuable and helpful in revising and improving our thesis, as well as providing important guidance for our research. We have carefully studied these comments and made revisions, which we hope will be accepted by you. Where changes have been made, we have highlighted them in yellow in the revised version.

We would love to thank you for allowing us to resubmit a revised copy of the manuscript and we highly appreciate your time and consideration.

Sincerely.

The main corrections in the paper and the responds to the reviewer’s comments are as flowing:

Q1.Introduction

  • More studies, aspects need in the introduction to show the main

research gap. 

  • The study goal needs more precise 
  • I suggest making hypotheses. Especially the Authors used an empirical model to their results, which could be also used as a hypothesis. 

Response:Thank you very much for your valuable comments. I have added relevant studies and hypotheses and enriched the research objectives in the introduction section.

Coach-athlete relationship has been linked to a wide range of variables connected to performance and athletes’ wellbeing, such as: affects, coping and sports satisfaction [1]. The 3C is a prominent model examining coach-athlete relationship and is mainly focused on the dynamic interrelations between the coach and athletes and their mutual influence [2]. However, the model was updated to the 3 + 1Cs, which also evaluated the perception between athletes and coaches [3].

To date, only a few studies examined the impact of the CAR on athletes’ affects [19]. This is particularly rare due to the possible connection between these variables and the impact of affective states on sports performance [20]and the increase in the literature on emotional contagion and interpersonal emotional regulation. Nevertheless, previous studies from other theoretical backgrounds close to CAR (e.g., coach behaviours, coach leadership) that have examined this relationship. Particularly, previous works [21]revealed that warming and supportive coaches were significantly related to athletes’ positive affects . Otherwise, unsupportive coaches were significantly related to athletes’ negative affects (NA) [22].

Based on the previous literature [27], we hypothesised that: (a) The coach-athlete relationship perceived by student-athletes had a direct positive and significant effect on athletic behaviours; (b) Psychological needs had a mediating effect between the coach-athlete relationship and athletic behaviours; (c) By improving the coach-athlete relationship, it may help to improve the psychological needs of college athletes.

Q2.Methods

  • Consider moving procedure to the first subchapter. In this case the readers were understand the data collection in the beginning of the chapter. 
  • Furthermore, I suggest combining participants with procedure ("Participants and Procedure)
  • The measures are well detailed. However, I suggest to put your primary analysis to the results and only make clear descriptions in this study on the measures. 
  • As I understand the Authors used exploratory (EFA) and confirmatory factor analysis (CFA) in this study. However, these are well known measures, and it is unnecessary to use both in these cases (Usually EFA and CFA are conducted on different sample). So, I encourage to only use CFA and move them to the results. 
  • Statistical analysis must be more detailed. For example, what was the dependent variables on step wise regression; What kind of fit indices was used for CFA; etc.

Response:Thank you very much for your valuable comments. I've moved procedure to the first subchapter and combining participants with procedure.The redundant EFA form has been removed and the CFA form has been supplemented with more detailed instructions.

Prior to data collection, our research team reviewed the questionnaire with three sport psychology PhDs. We compiled a list of athletic department directors to be surveyed and communicated with them via telephone or in-person visits to explain the study's purpose and request assistance with data collection. Based on each school's survey schedule, we provided participating athletes with a detailed explanation of the study's purpose and instructions for completing the questionnaire, which was distributed via QR code. We adhered to the relevant Research Ethics Committee guidelines for projects involving human participants, and all participants provided informed consent. Using a convenience sampling method, we selected seven colleges and universities from Zhejiang Province and Anhui Province, surveying a total of 260 college athletes. After excluding six invalid questionnaires, we analyzed 254 valid questionnaires, resulting in a validity rate of 97.69%. The sample included 113 female students (44.49%) and 141 male students (55.51%); 77 students specializing in football (30.31%), 55 in track and field (21.65%), and 26 in basketball (10.24%); 36 freshmen (14.17%), 42 sophomores (16.54%), 22 juniors (8.66%), and 154 seniors (60.63%).

2.3 Confirmatory factor analysis (CFA)

First, the normality of the sample was tested. The P-P plot indicated that the data points were distributed close to the straight line, with a maximum skewness of 0.901 and a maximum kurtosis of 0.887, suggesting that the data follows a normal distribution. Factor analysis was conducted using the maximum likelihood method to estimate the model. A confirmatory factor analysis (CFA) was performed on the three-factor structure model, and the results indicated a good fit. A comparison of the fit indices for the single-factor model and the three-factor model is presented in Table 2. The combined fit indices demonstrate that both the single-factor model and the three-factor model of the original questionnaire are acceptable.

Q3.Results

  • Line 191: should be "Descriptive statistic"
  • Table 4: Why were these measures used? M; SD; Skewness and Kurtosis could be enough, but reliability measures can be also added to this table. 
  • Correlation table seems to have missing values.
  • What was the model fit for figure 1, as it was mentioned it is a path analysis. 

Response:Thank you very much for your valuable comments. Line 191 has been amended to Descriptive statistic.

Q4.The credibility of the results is also questionable, as possible covariates have not been adjusted.

Response:Thank you very much for your valuable comments. We agree with you very much. The amendments have been made in accordance with the comments you have made. The selection of variables in Table 4 has been added in the text. the model fit for figure 1 was analysed in 2.3Confirmatory factor analysis (CFA).

The mean and standard deviation of the measured coach-athlete relationship, psychological needs, and motor behavior sub-factors were calculated in this study. Table 3 presents the results of the descriptive statistical analyses of these sub-factors. The analyses indicated that respondents perceived the highest levels of complementarity and closeness but lower levels of dedication in their relationships with their coaches. Regarding psychological needs, relationship needs were higher than autonomy needs. "Coach-Athlete Relationships," "Intrinsic," and "Motor Behavior" exhibited high variances and larger SEM values, indicating greater variability. In contrast, variables such as "Complementarity," "Autonomy," and "Sense of Capability" showed lower variances and smaller SEM values, suggesting less variability and more precise estimates. Finally, for motor behavior, intrinsic motivation was the highest, while extrinsic motivation levels were the lowest, indicating overall higher levels of intrinsic motivation.

Q5.Discussion, conclusion

  • This part needs to revise after revising the methods and the results. However the structure and main point of this part are well established. 

Response:Thank you very much for your valuable comments. I have revised the results and discussion sections.

The purpose of this study was to explore the associations between coach-athlete relationships, psychological needs, and motor behaviors of college athletes. The study revealed the following conclusions: (1) The coach-athlete relationship significantly positively affects psychological needs and athletic behavior. (2) In the context of the coach-athlete relationship, closeness had the most significant effect on motor behavior, followed by complementarity and dedication. (3) Psychological needs mediate the relationship between the coach-athlete relationship and motor behavior, emphasizing their role as a bridge connecting these variables. A strong coach-athlete relationship enhances athletes' psychological needs satisfaction, thereby promoting more active participation in motor behaviors. Future research should further explore the mechanisms by which different types of coach-athlete relationships affect psychological needs and motor behaviors, and design appropriate interventions to promote good relationship building and positive development of motor behaviors. These findings have important theoretical and practical implications for fostering a healthy and positive sports culture and improving overall athlete performance.

Reviewer 3 Report

Comments and Suggestions for Authors

There are several major points that I would like the authors should consider to improve in this manuscript. 

- Weak literature review: Need strong theoretical approach to justify the relationships in this manuscript rather than just previous findings of the proposed relationship.   

- In the introduction/literature, some comments or information (a scale validation for CART-Q) regarding China are necessary? If so, why? 

- The authors used all validated scales. All results and tables from EFAs should have not been included in this article. Just to conduct one CFA, including all variables to validify. 

- Pleas use SEM other than regressions. 

- After all new results, the authors should rewrite all results and discussions with more practical implications. 

Comments on the Quality of English Language

I believe the quality of English is big issue in this manuscript. 

Author Response

Dear Reviewer:

Thank you for your letter and for the reviewers’ comments concerning our manuscript. All these comments are valuable and helpful in revising and improving our thesis, as well as providing important guidance for our research. We have carefully studied these comments and made revisions, which we hope will be accepted by you. Where changes have been made, we have highlighted them in yellow in the revised version.

We would love to thank you for allowing us to resubmit a revised copy of the manuscript and we highly appreciate your time and consideration.

Sincerely.

The main corrections in the paper and the responds to the reviewer’s comments are as flowing:

Reviewer:

Q1.Weak literature review: Need strong theoretical approach to justify the relationships in this manuscript rather than just previous findings of the proposed relationship.   

Response:Thank you for your thoughtful and encouraging review of our study. I have added relevant research in the introduction section.

Coach-athlete relationship has been linked to a wide range of variables connected to performance and athletes’ wellbeing, such as: affects, coping and sports satisfaction [1]. The 3C is a prominent model examining coach-athlete relationship and is mainly focused on the dynamic interrelations between the coach and athletes and their mutual influence [2]. However, the model was updated to the 3 + 1Cs, which also evaluated the perception between athletes and coaches [3].

To date, only a few studies examined the impact of the CAR on athletes’ affects [19]. This is particularly rare due to the possible connection between these variables and the impact of affective states on sports performance [20]and the increase in the literature on emotional contagion and interpersonal emotional regulation. Nevertheless, previous studies from other theoretical backgrounds close to CAR (e.g., coach behaviours, coach leadership) that have examined this relationship. Particularly, previous works [21]revealed that warming and supportive coaches were significantly related to athletes’ positive affects . Otherwise, unsupportive coaches were significantly related to athletes’ negative affects (NA) [22].

Q2.In the introduction/literature, some comments or information (a scale validation for CART-Q) regarding China are necessary? If so, why? 

Response:Thank you for your insightful suggestions. It is indeed necessary to include comments or information regarding the scale validation for CART-Q in China. This is because linguistic differences in the translation of foreign literature can introduce errors, and the reliability and validity of questionnaire results can vary significantly across different ethnic groups. Based on your valuable feedback, I have revised the content of the paper to address these concerns.

Q3.The authors used all validated scales. All results and tables from EFAs should have not been included in this article. Just to conduct one CFA, including all variables to validify. 

Response:Thank you very much for your valuable comments. We agree with you very much. The amendments have been made in accordance with the comments you have made. The selection of variables in Table 4 has been added in the text. the model fit for figure 1 was analysed in 2.3Confirmatory factor analysis (CFA).

Table 2 Results of the confirmatory factor analysis

χ2/df

CFI

TLI

SRMR

RMESA

Model1

4.922***

0.683

0.667

0.103

0.109

Model2

2.791***

0.856

0.849

0.090

0.074

Model 1 is a one-factor model and Model 2 is a three-factor model. ***p<0.001

Q4.Pleas use SEM other than regressions. 

Response:Thank you very much for your valuable comments. I have added paths and tests for SEM modelling to the text.

In this study, MPLUS 8.0 was used to conduct a validation factor analysis on the structural validation sample data (n=254), which verified a good model fit of the coach-athlete relationship on sport behaviour, and a path analysis was conducted to estimate the path coefficients of the structural model and assess the mediating effect of psychological demands by SEM, and the results are shown in Table 5.

Table 5 Path analysis and mediated effects results

Path relationship

β

SE

z

Bootstrap 95%CI

p

Lower

Upper

Coach-Athlete Relationships → Psychological needs

0.50

0.05

9.00

0.35

0.55

<0.001

Psychological needs → Motor Behaviour

0.35

0.07

4.29

0.16

0.44

<0.001

Coach-Athlete Relationships → Motor Behaviour

0.25

0.06

3.33

0.08

0.32

<0.001

Coach-Athlete Relationships → Psychological needs → Motor Behaviour

0.175

0.04

3.50

0.06

0.22

<0.001

Q5.After all new results, the authors should rewrite all results and discussions with more practical implications. 

Response:Thank you very much for your valuable comments. After adding and revising the tables and data for the paper, I revised the results and conclusion sections.

The purpose of this study was to explore the associations between coach-athlete relationships, psychological needs, and motor behaviors of college athletes. The study revealed the following conclusions: (1) The coach-athlete relationship significantly positively affects psychological needs and athletic behavior. (2) In the context of the coach-athlete relationship, closeness had the most significant effect on motor behavior, followed by complementarity and dedication. (3) Psychological needs mediate the relationship between the coach-athlete relationship and motor behavior, emphasizing their role as a bridge connecting these variables. A strong coach-athlete relationship enhances athletes' psychological needs satisfaction, thereby promoting more active participation in motor behaviors. Future research should further explore the mechanisms by which different types of coach-athlete relationships affect psychological needs and motor behaviors, and design appropriate interventions to promote good relationship building and positive development of motor behaviors. These findings have important theoretical and practical implications for fostering a healthy and positive sports culture and improving overall athlete performance.

Round 2

Reviewer 2 Report

Comments and Suggestions for Authors

Thank you for the Author's response to my comments. The manuscript improved significantly, but it contains several methodological problems

·      Make the study goal and hypotheses a new paragraph. 

·      I suggest renaming "Research participants" to "Participants and Procedure"

·      Please put your primary analysis to the results and only make clear descriptions in this study on the measures.

·      Chapter 2.3 is not clear why it is necessary. Much of this information should be on statistical methods.  

·      I suggest using only skewness and kurtosis for normality measures.  

·      Chapter 2.4 should be the statistical analysis and add more information on CFA including used fit indices, measures, thresholds, etc. 

·      Table 4 should contain kurtosis and Cronbach alpha values instead of the measures. 

·      I suggest adding a correlation table where you correlate all your study variables since it will help you to explain path analysis more. 

·      Please describe why you use path analysis and regression models.  As I see it, path analysis would be enough since it could also be used to see the mediation effects. 

·      Here is my recommendation to structure results:

o   Primary analysis (includes CFA)

o Descriptive statistics and correlations

o   Path analysis or Regression

Comments on the Quality of English Language

Minor editing of English language required

Author Response

Dear Reviewer:

Thank you for your letter and for the reviewers’ comments concerning our manuscript. All these comments are valuable and helpful in revising and improving our thesis, as well as providing important guidance for our research. We have carefully studied these comments and made revisions, which we hope will be accepted by you. Where changes have been made, we have highlighted them in yellow in the revised version.

We would love to thank you for allowing us to resubmit a revised copy of the manuscript and we highly appreciate your time and consideration.

Sincerely.

The main corrections in the paper and the responds to the reviewer’s comments are as flowing:

Q1. Make the study goal and hypotheses a new paragraph.

Response:Thank you very much for your valuable comments.I have made the suggested modification by separating the study goal and hypotheses into a new paragraph. Your input has been instrumental in improving the clarity and structure of the manuscript.

Q2. I suggest renaming "Research participants" to "Participants and Procedure

Response:Thank you for your insightful suggestions. I have renamed the section "Research Participants" to "Participants and Procedure" as you recommended.

Q3.Please put your primary analysis to the results and only make clear descriptions in this study on the measures.

Response:Thank you very much for your valuable comments. I understand the importance of focusing on the primary analysis in the results section and providing clear descriptions of the measures. Given the critical importance of validating the questionnaire, I have simplified the descriptions as much as possible. Additionally, I have made the necessary revisions to the results section to better align with your suggestions.

Q4.Chapter 2.3 is not clear why it is necessary. Much of this information should be on statistical methods.

Response:Thank you very much for your valuable comments. I have revised Chapter 2.3 to clearly articulate its necessity and have relocated the relevant information to the section on statistical methods.

Confirmatory Factor Analysis (CFA) was conducted to validate the factor structure of the questionnaire used in this study. Prior to analysis, the normality of the sample distribution was assessed using a P-P plot, which indicated a distribution close to a straight line, with skewness up to 0.901 and kurtosis up to 0.887, suggesting adherence to normality assumptions. Factor estimation employed the maximum likelihood method.A CFA was then performed on both a one-factor and a three-factor structure model. The fit indices for each model were compared (see Table 2). Model 1, representing a single-factor structure, showed χ2/df = 4.922, CFI = 0.683, TLI = 0.667, SRMR = 0.103, and RMSEA = 0.109. Model 2, representing a three-factor structure, demonstrated improved fit with χ2/df = 2.791, CFI = 0.856, TLI = 0.849, SRMR = 0.090, and RMSEA = 0.074. The significance level indicated by ***p<0.001.These results underscore the adequacy of both the single-factor and three-factor models for interpreting the original questionnaire. This analysis provides a robust validation of the factor structure essential for subsequent statistical and substantive interpretations in this study.

Q5.I suggest using only skewness and kurtosis for normality measures.Table 4 should contain kurtosis and Cronbach alpha values instead of the measures.

Response:Thank you very much for your valuable comments. I have revised the manuscript to use skewness and kurtosis for normality measures as you suggested. Additionally, I have updated Table 4 to include Cronbach's alpha values instead of the previous measures.

Table 4 Descriptive statistics of measured variables(n=254)

Variable

M

SD

Kurtosis

Cronbach’s Alpha

Skewness

Intimacy

19.37

2.861

15.070

0.947

-3.307

Complementarity

19.32

2.156

1.964

0.951

-1.1278

Dedication

18.38

2.694

3.824

0.922

-1.711

Coach-Athlete Relationships

57.07

7.067

9.045

0.943

-2.448

Relevance

18.89

3.302

-0.881

0.836

0.183

Autonomy

12.46

2.023

0.370

0.887

-0.705

Sense of capability

18.38

2.255

5.175

0.851

-1.818

Psychological needs

49.73

6.065

2.363

0.853

-0.807

Intrinsic

69.30

9.084

6.020

0.951

-2.067

Identified

55.18

6.699

5.785

0.963

-1.844

External

15.33

3.764

-1.009

0.969

0.089

Motor Behaviour

139.81

17.004

4.951

0.958

-1.583

Q6.Chapter 2.4 should be the statistical analysis and add more information on CFA including used fit indices, measures, thresholds, etc.

Response:Thank you very much for your valuable comments. We agree with you very much. I have revised Chapter 2.4 to focus on statistical analysis and have added more detailed information on Confirmatory Factor Analysis (CFA), including the fit indices used, measures, and thresholds.

The collected data underwent rigorous analysis using SPSS 25.0 and MPLUS 8.0 software, aligning with the study's objectives. Initially, exploratory factor analysis (EFA) and internal consistency analysis were employed, followed by confirmatory factor analysis (CFA), to establish the validity and reliability of each questionnaire's factor structure. Normality testing of the sample distribution preceded the factor analysis, confirming that the data approximated a normal distribution. The maximum likelihood method was utilized for factor estimation. The CFA was subsequently applied to evaluate a three-factor structure model, which demonstrated a satisfactory fit based on the analysis results.Furthermore, descriptive statistics and correlation analyses were conducted to scrutinize the sub-factors derived from the validated measurement variables identified through factor analysis. To explore the influence of these variables, stepwise multiple regression analyses were performed. Lastly, path analyses and mediation effects tests were executed to validate the hypothesized relationships within the research model. These comprehensive analytical steps were crucial for assessing the data and interpreting findings within the context of the study's research objectives.

Q7. I suggest adding a correlation table where you correlate all your study variables since it will help you to explain path analysis more.

Response:Thank you for your insightful suggestion. I have added a correlation table that includes all the study variables, which will help in explaining the path analysis more effectively.

 Table 5 Bivariate correlations of all study variables

Variable

1

2

3

4

5

6

7

8

9

10

11

12

1.Intimacy

1

2.Complementarity

0.800***

1

3.Dedication

0.743***

0.735***

1

4.Coach-Athlete Relationships

0.932***

0.909***

0.906***

1

5.Relevance

0.271***

0.273***

0.432***

0.357***

1

6.Autonomy

0.271***

0.249***

0.141*

0.239***

0.512***

1

7.Sense of capability

0.738***

0.663***

0.641***

0.745***

0.351***

0.510***

1

8.Psychological needs

0.512***

0.478***

0.520***

0.551***

0.845***

0.802***

0.773***

1

9.Intrinsic

0.509***

0.414***

0.363***

0.471***

0.261***

0.470***

0.389***

0.443***

1

10.Identified

0.446***

0.427***

0.383***

0.457***

0.458***

0.628***

0.487***

0.640***

0.758***

1

11.External

0.269***

0.468***

0.292***

0.363***

0.428***

0.303***

0.319***

0.453***

0.401***

0.553***

1

12.Motor Behaviour

0.507***

0.493***

0.410***

0.512***

0.415***

0.566***

0.471***

0.589***

0.992***

0.921***

0.645***

1

*P<0.05,**P<0.01,***P<0.001

Q8.  Please describe why you use path analysis and regression models.  As I see it, path analysis would be enough since it could also be used to see the mediation effects.

Response:Thank you for your thoughtful and encouraging review of our study.We employ both path analysis and regression models to ensure a more comprehensive and nuanced understanding of our study's results. Regression models are instrumental in validating individual relationships between variables. This step is essential to confirm that the relationships included in the path analysis are significant and reasonable. Moreover, regression models help simplify complex models and test important assumptions such as linearity, normality of residuals, and homoscedasticity. The combined use of these methods enhances the robustness and interpretability of our findings, enabling us to draw more accurate and holistic conclusions.

Reviewer 3 Report

Comments and Suggestions for Authors

I am fine with the revisions that the authors have made. 

Comments on the Quality of English Language

NA

Author Response

Thank you for your review and for accepting the revisions we have made. We appreciate your constructive feedback and support throughout the review process.

Round 3

Reviewer 2 Report

Comments and Suggestions for Authors

Thank you for the Author's response to my comments. The manuscript improved significantly, but it still has methodological problems

·      Please use a "primary analysis" in your results and move all your CFA models to this new section. The main problem is that it's hard to follow and we don't know what fit indices were used. 

·      Please describe fit indices with the cut of points in the statistical methods part.

·      It's unclear why the Authors added a CFA subchapter (2.3) while CFA was used before in all the measures. It's not clear what model was tested as well. 

·      I believe restructuring the methods and the results would be important to understand this study. 

Comments on the Quality of English Language

 Minor editing of English language required

Author Response

Dear Reviewer:

Thank you for your letter and for the reviewers’ comments concerning our manuscript. All these comments are valuable and helpful in revising and improving our thesis, as well as providing important guidance for our research. We have carefully studied these comments and made revisions, which we hope will be accepted by you.

We would love to thank you for allowing us to resubmit a revised copy of the manuscript and we highly appreciate your time and consideration.

Sincerely.

The main corrections in the paper and the responds to the reviewer’s comments are as flowing:

Q1. Please use a "primary analysis" in your results and move all your CFA models to this new section. The main problem is that it's hard to follow and we don't know what fit indices were used. 

Response:Thank you very much for your valuable comments.I have added the CFA section to 3.1 Primary Analysis, and the details about the fit indices are mainly included in section 2.2.

Q2. Please describe fit indices with the cut of points in the statistical methods part.

Response:Thank you for your insightful suggestions. I have supplemented the fit indices in section 2.2 Measurement Tools.

Q3.It's unclear why the Authors added a CFA subchapter (2.3) while CFA was used before in all the measures. It's not clear what model was tested as well. 

Response:Thank you very much for your valuable comments. The CFA section was added upon the recommendation of another reviewer.Confirmatory Factor Analysis (CFA) was conducted to validate the factor structure of the questionnaire used in this study. Factor estimation employed the maximum likelihood method.A CFA was then performed on both a one-factor and a three-factor structure model. The fit indices for each model were compared (see Table 2). Model 1, representing a single-factor structure.These results underscore the adequacy of both the single-factor and three-factor models for interpreting the original questionnaire. This analysis provides a robust validation of the factor structure essential for subsequent statistical and substantive interpretations in this study.

Q4.I believe restructuring the methods and the results would be important to understand this study. 

Response:Thank you very much for your valuable comments. I have revised the results section to the following format: 1. Primary Analysis (includes CFA) 2. Descriptive Statistics and Correlations 3. Path Analysis.